# An Ion Source's View of Its Plasma

Peter Spädtke 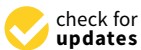

Ingenieurbüro für Naturwissenschaft und Programmentwicklung, Junkernstr. 99, 65205 Wiesbaden, Germany; p.spaedtke@inp-dme.de

**Abstract:** Modeling of ion beam extraction from an ECRIS requires special procedures in order to achieve results similar to what is found experimentally. The initial plasma conditions must be included for consistency between experiment and simulation. Space charge forces and their compensation of the extracted ion beam become important with increasing beam intensity. Here we consider the various beam-plasma conditions that occur along any beam line.

**Keywords:** ion beam; extraction; space charge



## 1. Introduction

High quality ion beams are required in a growing number of fundamental and technological areas, and a cold plasma from which the ion beam can be formed is a necessary prerequisite. By "high quality" it is meant that all particles within the assemblage behave similarly to each other, implying in turn that their temperature is low. This is equivalent to the requirement of having a high density in six-dimensional phase space, given by three spatial coordinates and three momentum coordinates. According to Liouville it can be shown that under certain conditions (e.g., only conservative forces act) this 6D phase space volume remains constant.

Many different types of ion sources have been developed and are available, and good overviews have been published [1,2]. Still, each ion source has its own set of specific working conditions and therefore plasma properties. Some of these unique peculiarities have to be considered when computer simulations are used to model ion beam extraction from the ion source and the subsequent beam transport.

Computer codes have been developed to model the behavior of charged particles in electric and magnetic fields. They are used to optimize the source extraction geometry, to preserve the quality of the cold plasma, and to not decrease the beam quality by aberrations due to nonlinear optics or similar effects.

In the case of a plasma where collisions (not a conservative force) ensure a homogeneous plasma density distribution near the extraction aperture, Child's Law describes the matching condition with $j = \frac{4}{9}\pi\epsilon_0\sqrt{q/m}\Delta\Phi^{\frac{3}{2}}/d^2$, where j is the current density, $\epsilon_0$ is the permittivity of free space, $m$ the mass and $q$ the charge of the ion, and $\Delta\Phi$ is the potential difference across the acceleration gap $d$. This matching condition describes the equilibrium between plasma pressure and extraction field strength to achieve the best possible beam. If the plasma density is too low, the ion beam will be overfocused and vice versa if the plasma density is too high the ion beam becomes divergent due to high space charge forces. In between both extreme densities the ion beam has its highest phase space density. This condition is called perveance match.

If magnetic fields are not negligible, the plasma is anisotropic, and Child's Law is not necessarily fulfilled everywhere over the extraction aperture region. Local differences between emission-limited and space-charge-limited flow can occur.

Whereas in the magnetic field-free case the six dimensions can be separated to fewer dimensions without loss of generality, this is not true anymore in the magnetic case due to the coupling of different subspaces. Coupling between different dimensions of this 6D

phase space might also take place, for example, if space charge forces are relevant or if collisions between particles become important.

The initial conditions of the plasma particles will determine the beam quality: the initial plasma temperature will be conserved of course, and the absolute phase space density will shrink due to the acceleration. Ions starting within a solenoidal field will execute skewed trajectories according to Busch's theorem. Some ion sources use additional magnetic fields, which may well introduce even more aberrations. Such dependencies need to be taken into account when simulating a system. Plasma physics plays an important role for a correct description of any of these ion optical systems.

## 2. The Ion Beam Source

In this article, the application of computer simulation will be demonstrated for the special case of an Electron Cyclotron Resonance Ion Source (ECRIS) and a recipe will be given for the simulation procedure for this specific kind of ion source. An ECRIS can best be described as a combination of two different machines: an electron accelerator to provide high energy electrons that are capable of stepwise ionizing heavy ions to higher charge states and storage for those ions inside a plasma chamber (PC) until a certain ion charge state distribution (CSD) has been achieved; this is then followed by extraction of these ions from the ion source for their intended application.

One version of such an ECRIS is CAPRICE [3], shown in Figure 1. This ion source was developed from a precursor device used in early fusion experiments, with which it was shown that the ion particle density $n_i$ necessary for fusion of Deuterium and Tritium ($^2De + {}^3T \rightarrow {}^4He + n + 17.6\,\text{MeV}$) could not be achieved efficiently enough with a simple mirror device. Additional radial confinement force seemed to be necessary to increase the plasma density. By increasing the radial confinement by means of an additional hexapole magnetic field colinear with the magnetic mirror field, the particle densities were increased, but the numbers were still not adequate for fusion, and the worldwide controlled fusion research effort experiments went on to other devices such as torus-shaped stellarators and tokamaks.

Geller [4] recognized the potential of this improved mirror device for application as an ion source, especially for higher charge states of heavy ions. Technical advances in magnet technology (superconductivity) and high frequency power generation (GHz technology in the kW range and up) made this type of ion source possible and attractive. Particle accelerators could greatly benefit from the availability of higher charge states of heavy ions. The use of higher charge states in an accelerator means higher velocity gain for the same potential drop across the acceleration gap. The lower the $m/q$ ratio (mass-to-charge), the higher the velocity of the ions and the more simply (shorter) an rf accelerator can be built. In most cases, the ion velocity has to be matched to the geometry and frequency of the following rf-accelerator.

Since the very beginning of ECRIS development, effort has been directed to improvements toward higher intensities for higher charge states and the complex dependencies of this multiparameter field on the plasma properties and their impact on beam quality have been clarified. Some of these are listed in the following.

The dependency of the intensity and CSD of the extracted ion beam on frequency and magnetic field strength of the ECRIS device was one of the first important features to be observed and clarified, but there are other parameters that also influence the particle density in the plasma from which the ion beam can be extracted. Even small frequency shifts of the heating power have a strong influence on the extracted beam current. These frequency shifts are so small that their effect cannot be explained via changes in the electron cyclotron resonance condition ($\omega_{ce}\,[\text{GHz}] = 28 \cdot B\,[\text{T}]$). Mixing different frequencies to heat the plasma shows similar effects. Adding various isotopes to the plasma might influence the extractable ion beam current, as does providing additional electrons to the plasma (for example, by using biased probes). The volume of the PC might play a role, via the power density within this volume. Furthermore, there are more dependencies. Different theories

have been proposed to explain the influence of each parametric variation, but up to now no complete theory has been developed to describe the generation and confinement of the ECRIS plasma.

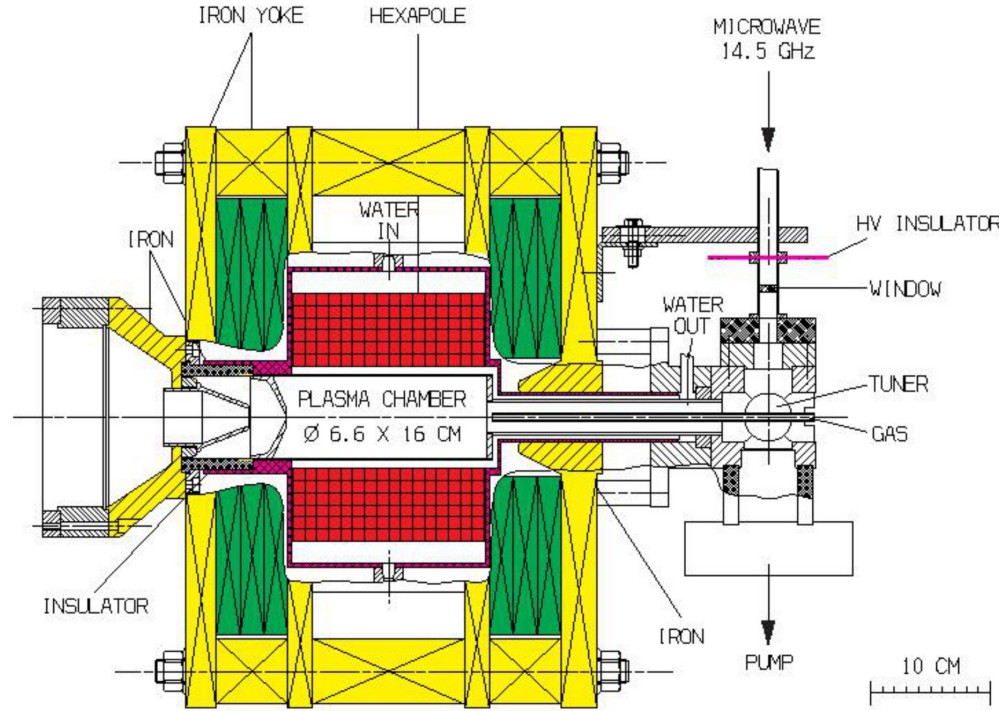

**Figure 1.** The Caprice ECRIS. Normal conducting coils (shown in green) together with a permanent magnet hexapole (shown in red). Microwave power is fed to the source from a high power generator at 14.5 GHz, connected by waveguide to the ECRIS. Iron parts in yellow. Gas is injected from the side opposite to the extraction side.

Besides these unknown dependencies, from a mathematical point of view two different types of differential equations have to be solved when simulation programs are used: boundary values are required for the solution of the equations for the electromagnetic potential (Laplace and Poisson) and solving the trajectories is an initial value problem for ray tracing. For the first case a mesh is used on which the differential equation (DE) can be solved, whereas the 2nd order DE for ray tracing can be transformed and integrated twice, using these calculated fields.

### 3. Ion Beam Extraction

Computer simulation can contribute to the optimization of a technical system as well as the experimental approach. Simulation codes for charged particles in electromagnetic fields were developed as early as the 1970s, initially for electrons emitted from a thermionic cathode [5]. Simulation codes followed shortly for ions emitted from a plasma with an assumed homogeneous density distribution [6]. With improvement of computer hardware (available memory, CPU speed), 3D simulation codes followed one decade later [7]. With such 3D capabilities, simulation codes could handle the physics of ion beam extraction and beam formation even in the presence of arbitrary electric and magnetic fields; the absence of any time dependency saves much computing power, otherwise a PIC (particle in cell) code would be required. However, the extraction of an ion beam from an ECRIS is notably different from extraction from classical magnetic field-free ion sources, and a special recipe has to be used to obtain realistic results from simulations.

For the solution of Laplace's equation ($\Delta\Phi = 0$), boundary conditions for the electric potential $\Phi$ on the surface of the enclosed volume are required, and for the solution of Poisson's equation ($\Delta\Phi = \rho/\varepsilon_0$) the space charge $\rho$ (if not negligible) also needs to be

taken into account. The space charge $\rho$ derives from positive (and/or negative) ions and negative electrons as $\rho = \rho_i + \rho_e$. The electron density $n_e$ below and near the plasma potential depends on the potential $\Phi$ and its deviation from the plasma potential $\Phi_{pl}$:
$n_e = n_{e_0} e^{-\frac{\Phi_{pl}-\Phi}{k_B T_e}}$, where $n_{e_0}$ is the electron density in the undisturbed plasma, $T_e$ is the electron temperature and $k_B$ Boltzmann's constant. At plasma potential, charge neutrality $n_i = n_e$ is assumed.

A plasma boundary between undisturbed plasma and extracted beam will develop where the electron density drops, according to the above equation. When discussing the value of $T_e$ in ECRISs it is important to distinguish two separate groups of electrons; they have very different energies. Some plasma electrons are accelerated by the resonant (to the local electron cyclotron frequency) microwave power and execute closed orbits determined by the magnetic field shape; these electrons can attain very high energy. It is common that a high X-ray flux is present in the near vicinity of an ECRIS, indicating the presence of electrons with energy up to MeV [8]. Coexisting with this group of hot electrons is a population of low energy electrons that have not been involved in the resonant heating process and that serve to maintain charge neutrality of the plasma.

The power for sustaining a discharge is provided by microwave (state of the art: 10 to 60 GHz) fed by an rf wave guide to the PC (not described in detail here).

Atoms need to be introduced to the discharge chamber where they can be ionized and further stripped to higher charge states. For this purpose, gas may be bled into the device if the desired species exists in gaseous form or for nonvolatile materials, sputtering, oven technologies, or chemical methods might be used. Which technology is employed depends on the specific element.

The plasma ions have a certain initial velocity distribution, depending on their temperature. The initial ion temperature depends on the plasma generation process and also on the ion and electron densities $n_i$ and $n_e$. If no plasma simulation code is used to determine the ion velocity distribution, the initial distribution needs to be defined by the simulation software user according to theoretical considerations.

In-experiment collisions may serve to ensure that the particle density distribution immediately in front of the extraction aperture is homogeneous and isotropic. However, if magnetic fields are present and are to be included in the simulation, we need to determine whether electrons or even ions are "magnetized"—whether the electron/ion Larmor radius $r_L = mv_\perp/\mathbf{B}$ is less than or greater than the characteristic geometric dimension of the PC, e.g., its diameter $\varnothing_{PC}$; here $v_\perp$ is the particle velocity perpendicular to the magnetic field. The mobility of electrons depends on the electron velocity and the magnetic field. In general, the electrons are magnetized ($r_{Le} < \varnothing_{PC}$), and as the magnetic field strength increases the ions may become magnetized, too. The transport of charged particles remains uncomplicated in the direction of the magnetic field but is limited perpendicular to the field; the plasma is not isotropic anymore.

By applying a strong electric field between the plasma and an appropriately designed "extractor system", ions from the plasma can be extracted (removed from the plasma) and accelerated by this field, whereas plasma electrons will be decelerated and reflected back into the plasma as long as their energy is lower than the potential drop within the extraction system. Only those ions whose gyro-center enters the extraction aperture can be extracted.

Even though optimization of this extractor system is a main application of computer simulation, it is not described in detail; just the general procedure is shown here. Because the ion's space charge influences its own path, a self-consistent solution must be found for the simulation. Such a problem can be solved by an iterative method: first the geometry needs to be transformed to a mesh, fine enough to describe the details, then solving Laplace's equation, calculating the path of ion trajectories using the **E**-field of Laplace's equation on this mesh, distributing the space charge $\rho$ to the mesh, and then solving Poisson's equation, executing ray tracing, and distributing space charge to the mesh again. Repeated execution of the last three steps should converge to a self-consistent solution if physically useful parameters are assumed.

However, the correct initial coordinates for the ions are still unknown at this point. The solution is to view the model from a hydrodynamics perspective in which the magnetic field lines cross the extraction aperture at different radii. These field lines show the possible trajectories on which ions can reach the extraction aperture and separate from the plasma as shown in Figure 2, where two projections (horizontal and vertical) of the three-dimensional magnetic field lines are shown. In this figure the color of each field line changes from yellow to red when the local magnetic flux density $B$ becomes stronger than $B_{ex}$ at the extraction plane. For ion beam simulation, the spatial coordinates of each of the magnetic field lines (or part) are used to define the starting locations for ions along that path (inside the plasma).

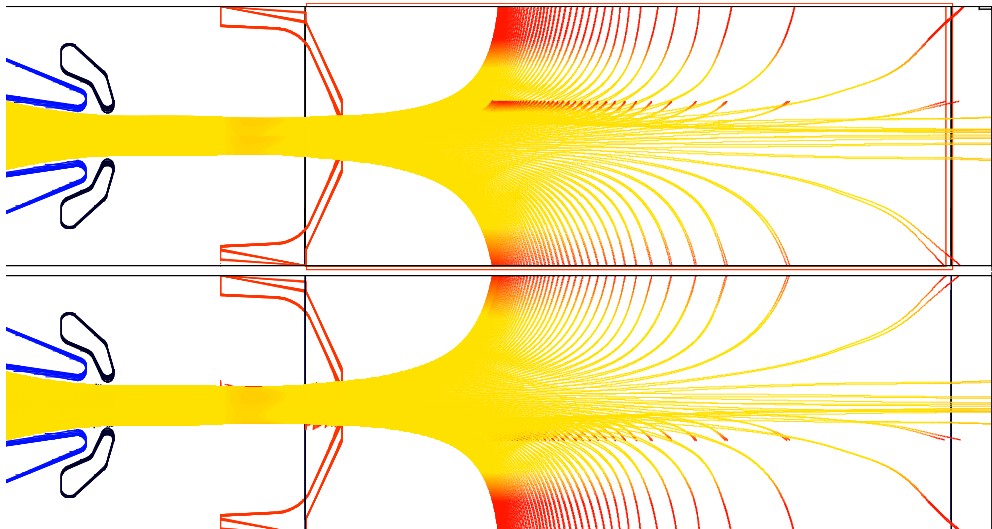

**Figure 2.** Projection of magnetic field lines crossing the extraction aperture on a radius of 4 mm. Top: horizontal, bottom: vertical. The field lines shown originate from one of three radial loss lines, depending on the azimuth. The field lines to the other three radial loss areas go the other (injection) side. Thus it is evident that the full length of the PC must be included in the simulation; this occupies about 70% of the total simulation volume. The plasma electrode (red) is opposite to a negative electrode (black) and a grounded electrode (blue). This kind of system is called an accel–decel extraction system.

This assumption has its experimental verification. As found in experiments performed at GSI Darmstadt, Figure 3 shows the erosion marks on the PC wall and on both end-plates of the PC (called injection side and extraction side), which occur only where magnetic field lines enter the material.

Two types of erosion marks can be identified in Figure 3. In the right-hand image a very thin crater can be identified, having some depth, and it is absolutely straight in radial direction. The other erosion mark is thicker in the azimuthal direction and shows some azimuth deviation from a straight line especially at larger radii. One possible explanation could be that one mark is caused by high energy electrons and the other mark by low energy plasma ions. Another interpretation would be to identify the straight line with direct ion losses instead of the broader erosion as caused by more collisional plasma ions when reflected and oscillating within the mirror. A third possible interpretation might be the action of the stray field from the hexapole. The correct clarification to the question has not yet been reported.

The initial ion velocity is given by the plasma temperature and should be distributed over the direction of the magnetic field line at this specific location.

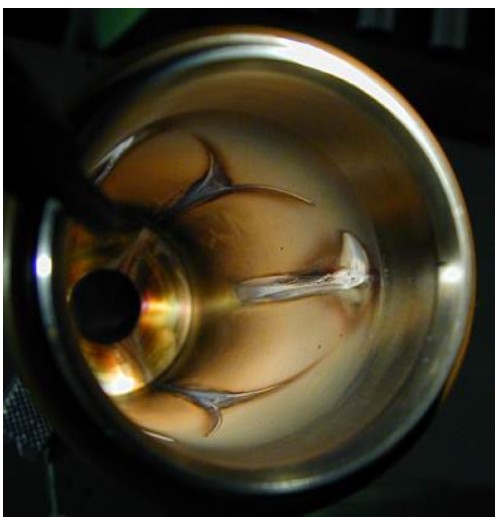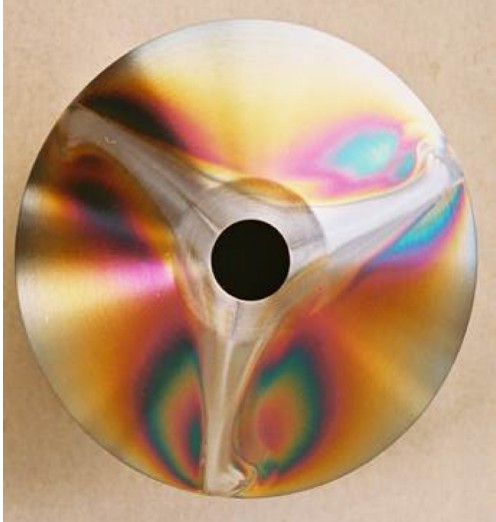

**Figure 3.** Erosion marks after some operating time. Left: Interior of the PC, showing the six loss lines and the region where the radial component of the hexapole field adds to the stray field of the mirror. Right: plasma electrode (plasma side) with extraction aperture, showing three of six loss lines. Note that these loss lines are generated by plasma losses.

Having defined the six phase space coordinates, only the intensity is still missing. In the simulation software used [9], this is an additional property for each ion trajectory instead of defining the electrical current by the number of trajectories. Along the trajectory, $div(\mathbf{j}) = \mathbf{0}$ is assumed. The individual current density of each trajectory can be constant but might be specified individually for each trajectory to allow an arbitrary plasma density distribution.

Along each magnetic field line, $grad(\mathbf{B})$ can be either positive or negative, either of which will transform longitudinal ion momentum into transverse ion momentum or vice versa, along the path of the guiding center, depending on the sign of $grad(\mathbf{B})$ (longitudinal and transverse are both with respect to the magnetic field line). Only that fraction of ions with enough energy to reach close enough to the extraction region can be extracted.

Different ion charge states are extracted and accelerated by the extraction system simultaneously. The experimental proof is shown in Figure 4, where a mixture of Argon and Helium is used to create a plasma from which an ion beam is extracted. The extracted ion beam is stopped on a viewing target (VT) located about 30 cm distant from the extraction system. A number of rings are visible, each ring representing a specific $m/q$. These rings are not perfectly round; it seems that they are constructed with three wings (120° symmetry).

Using a mean charge state in simulation instead of a real charge state spectrum will lead to incorrect results because the velocity gain for the different ion charge states is different, and the space charge does not scale linearly (beside of the obvious dependency caused by the magnetic field). Simulations confirm the different action of the magnetic stray field of the ion source on the different charge states.

A VT provides the profile (y-z) only, whereas different projections of the 6D phase space found by simulation provide more complete information. The projection of the 6D phase space into the y-y', respectively z-z' space is called emittance, the projection y-z', respectively z-y' shows the coupling. Furthermore, y and z are both transverse coordinates; y' and z' are the angles with respect to the longitudinal direction x. These definitions will be used later when simulation is discussed.

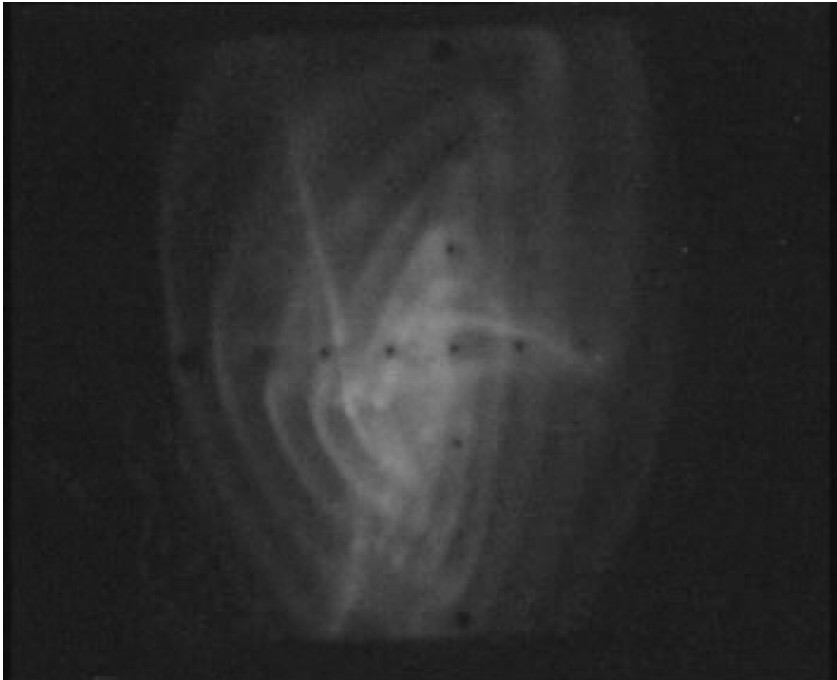

**Figure 4.** Photograph of ion beam profile on a VT 20 cm from the extraction system. The different "triangular rings" (or perhaps better, "wings") are caused by ions with different $m/q$ (Argon and Helium), guided differently in the stray magnetic field of the ion source. The outermost ring represents the lowest ion charge state of the operating gas, here Argon mixed with Helium, presumably $Ar^{2+}$, $m/q = 20$. With increasing charge state the ion velocity increases and the effect of the stray magnetic field becomes stronger. The lowest $m/q$ values (for $^1H^+$, $^1H_2^+$, $^4He^{2+}$, $^4He^+$) seem to be already overfocused at the VT location.

Figure 5 shows two different effects: first, the extracted ion beam from Figure 4 is focused by a beam line solenoid located 30 cm downstream onto a 2nd VT.

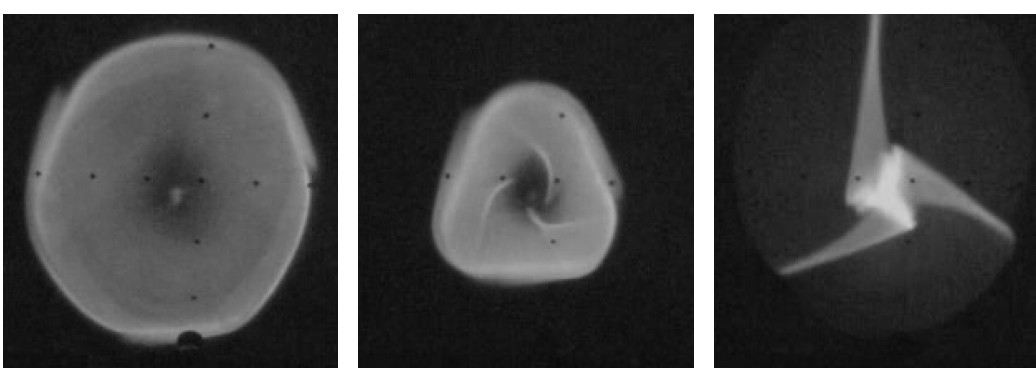

**Figure 5.** A magnetic solenoid behind ion beam extraction focuses the beam shown in Figure 4 onto a VT. Only singly charged Helium, $m/q = 4$ is visible.

The individual $m/q$ profiles of the beam look similar (perhaps identical?) when the lens is set according to this ratio. This is true also for the lowest possible $m/q$ of 1, thus precluding the assumption that overfocusing of higher charge states could be the reason for the specific optical effects observed with ion beams created by any ECRIS. An additional justification has been provided by using another type of ion source instead of an ECRIS. With a multicusp ion source at the ECRIS beam line, these effects could not be reproduced.

The second effect is identical for each $m/q$: A slight change in focusing strength of the beam line solenoid shows a series of different figures showing only one $m/q$ ($^4He^+$); the other $m/q$ values ($^4He^{2+}$, $^1H^+$, $^1H_2^+$) are already overfocused and are not visible anymore.

First, the beam is roughly round but remains hollow (left). With increasing focusing strength it becomes more triangular and still hollow. Three tips that start from the outside circumference of that triangle already touch the mid-axis of the hole (center). Further increase of the magnetic focusing strength causes these three tips to become overfocused. This part might have a larger angle and thus a lower longitudinal energy (right). This could explain the different focusing for different parts of the beam.

Figure 6 shows the $m/q$-analyzed fraction of the extracted ion beam behind a dipole magnet. Lower momenta are to the left.

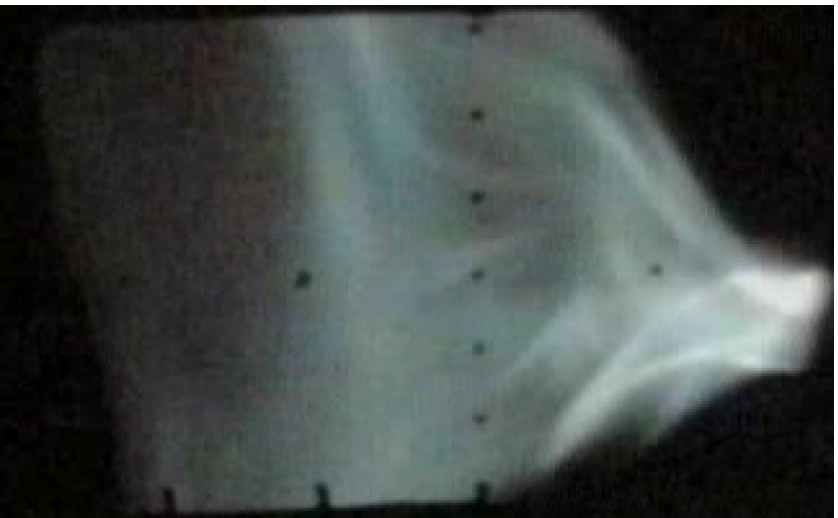

**Figure 6.** Profile of the beam from Figures 4 and 5 behind a dipole shows an even more complicated structure in the intensity distribution. Only one $m/q$ is visible here. Lower momenta appear on the left side. The question arises: how does one explain the visible structure? Are there three lines visible, or even more?

All observed profiles of the extracted ion beam depend on the magnetic field setting of the ion source [10]. In that publication the fact is essential that all $m/q$ develop the same profile when the focusing of the beam line optics is adjusted. What fraction of possible extraction paths is filled with ions inside the plasma might depend on frequency fine tuning [11]. In this way, the influence of fine tuning over the MHz range of 14.5 GHz microwave heating power has been demonstrated. This could be interpreted as the filling of available extraction field lines by modification of the local plasma density.

Different magnetic multipole configurations for the ion source (quadrupole, hexapole, octupole) have been tried in the past to modify the radial confinement, but no systematic comparison between the different systems is available with respect to beam quality. In the case of the quadrupole configuration, a slit for extraction seems to be favorable, compared to a round aperture, which seems to be better suited for higher order multipoles. The higher the order of the multipole the stronger and the more nonlinear the radial confinement.

Meanwhile, superconducting ECRISes have been taken into operation worldwide, e.g., in the US (LBNL, Berkeley) [12], in Japan (Riken, Tokyo) [13] and in China (IMP, Lanzhou) [14]. These versions use 28 GHz for plasma heating, but 45 GHz and 56 GHz are already under investigation. Recent developments in this field, using different magnetic field layouts and other additional new ideas, are the MARS [15] and the Cube [16] ion sources. Actually, the strength of the magnetic flux density is the limiting factor.

In general, the ion beam characteristics of any ECRIS are similar to one another. Of course, the effect of magnetization becomes stronger with increasing magnetic field strength, and simulation becomes increasingly important for understanding ion beam generation and ion beam extraction.

To achieve advantage from simulation, all relevant parameters need to be scanned. Because of this multidimensional parameter field this can become very data intensive. In

most cases a data reduction for diagnostic reason is useful. In Figure 7, only one charge state of the full spectrum has been selected, whereas in Figure 8 just a fraction of this is shown. Only the particles generated on magnetic field lines passing through the extraction aperture on a radius of 4 mm are displayed. The simulated ion trajectories are very close to the experimentally-found results. This is of course not a proof of theory but there is at least no contradiction to the assumed model.

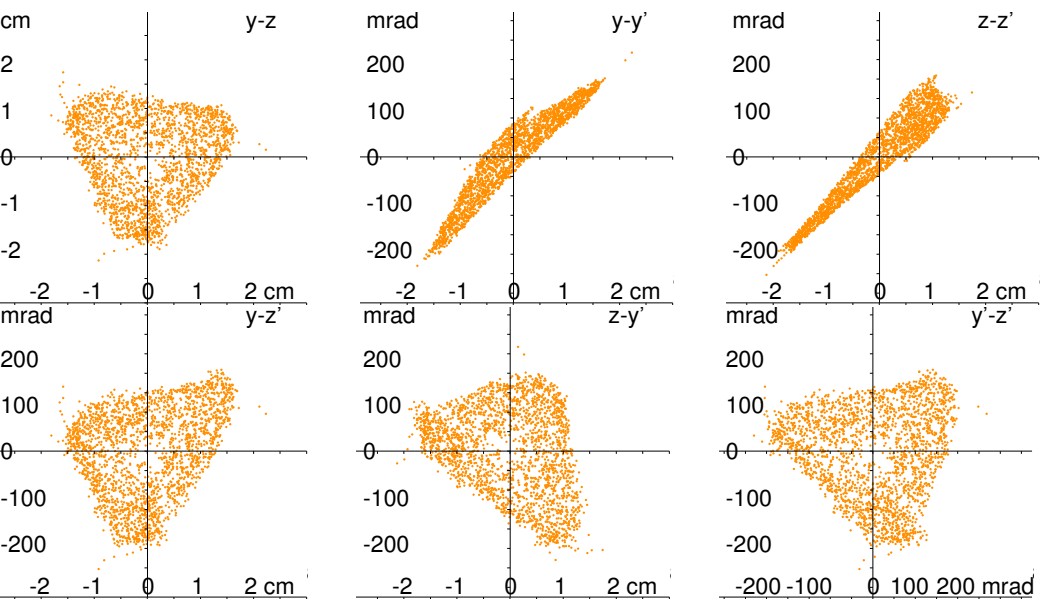

**Figure 7.** Different projections of the six-dimensional phase space. Top, from left to right: profile y-z, horizontal emittance y-y′, vertical emittance z-z′. Bottom, from left to right: y-z′, z-y′ and y′-z′, at x at the location of ground electrode, all $\dot{x}$. Only one $m/q$ ($^{40}$Ar$^{3+}$) is shown here. Simulation made for ECRIS CAPRICE.

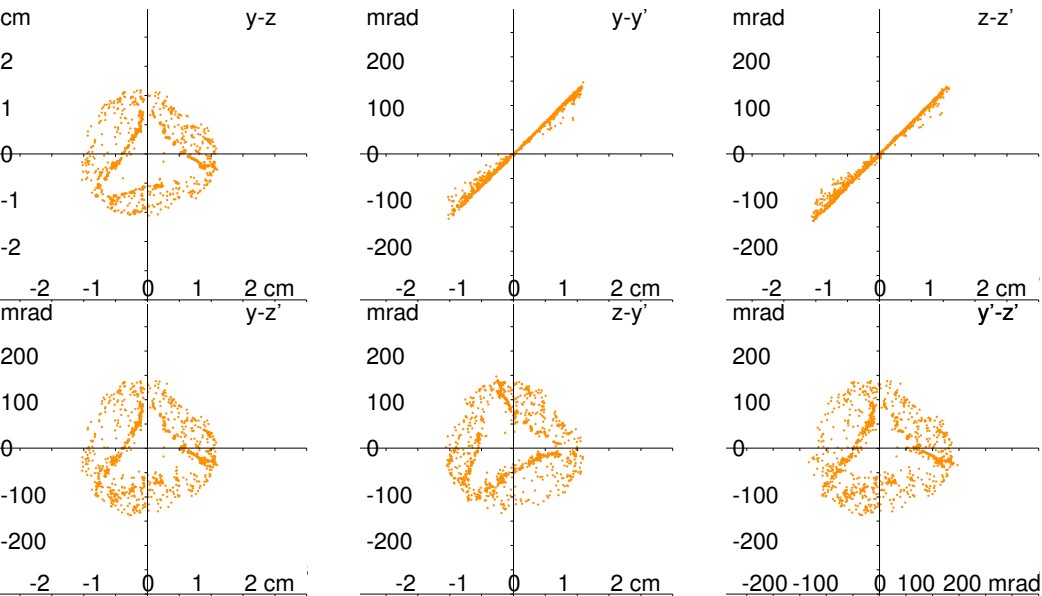

**Figure 8.** Different projections of the six-dimensional phase space. Same order as in Figure 7. Only one $m/q$ ($^{40}$Ar$^{3+}$) is shown here. Only these ions are shown, initiated on a magnetic field line which passed through the extraction aperture and only on a fixed radius of 4 mm. Simulation made for ECRIS HIISI [17].

Something that is not shown here but that is necessary to mention: using starting coordinates for the ions just in front of the extraction aperture results in a nice, round beam. This is in contradiction to the experiments. From that, the author's conclusion is that such starting conditions are not realistic. Using the initial coordinates for ions as described before, the simulated profiles are as seen in all experiments.

## 4. Ion Beam Transport

In the extraction system the ions are accelerated and experience external **E** and **B** forces as well as their own space charge (SC), which acts as a repelling force. While an electric force is relevant for ions, their own magnetic field is negligible at ion velocities ($\beta = v/c < 1\%$) gained by acceleration by the extraction voltage (and possible postacceleration) and electrical currents in the mA range.

Positively charged ions create a positive SC potential after extraction. Generally, no analytic solution for arbitrary ion beams exists, but for a rough estimate consider an infinitely long, circular "rod" of ions with a homogeneous charge distribution; the potential drop across such an ion beam can be estimated as $\Delta\Phi$ [V] = 30·$I$[A]/$\beta$ (nonrelativistic).

However, after extraction the conditions change; electrons might be produced by collisions of extracted ions with residual gas atoms or by any other process. Electrons with higher energy than the above-described potential drop caused by SC might escape from the ion beam. If only **E** fields originated by the ion beam itself are present, these electrons will oscillate within the ion beam. Only the coldest electrons will remain in the beam and with time the number of cold electrons will increase. These electrons might compensate the SC potential established by the positive ions via their negative charge. We have a fast-moving beam of positive ions and a cloud of negative electrons not necessarily moving together with the ions. This was observed already in the 1940s, when Uranium isotope separation was performed with ion beams [18]. For HIF (heavy ion fusion), more experiments have been done, especially for short ion pulses [19], avoiding SCC.

To avoid the ion source (at high positive potential) acting as an electron collector, in most cases an accel–decel extraction system is employed; in such a three-electrode system, the plasma electrode is at positive potential (as is the ion source itself), the intermediate electrode is at negative potential, and the outermost electrode is at ground potential. Electrons on the downstream side of the extractor see a negative potential (the negatively biased intermediate electrode) and are blocked from transport into the ion source. Such an accel–decel extraction system is necessary for any ion source working under high SC condition to avoid a strongly divergent ion beam. This is true also for the case when the ion beam is postaccelerated after extraction. Such an acceleration gap always requires a similar accel–decel system to avoid electron loss from the accelerated ion beam. A screening electrode (also called suppressor or accel electrode) should also be used directly in front of an rf-accelerator so as to preserve space charge neutrality [17].

There are exceptions, and they should be kept in mind. Extraction from some ion sources delivers high current ion beams that are clearly SCCed but without using an accel–decel system. A Penning ion source (PIG) source with radial extraction is one such, see Figure 9. Electrons are magnetized within a magnetic dipole field that is perpendicular to the extraction field **E**. The **B** field ensures that no electrons, required for space charge compensation, are lost due to the extraction field **E**.

In this context, the ion source is here described only by an annular anode located within an axial magnetic field with an extraction slit in the same axial direction. Ions of different $m/q$ ratios will be separated in the magnetic field during and after extraction. The ion energy is given by the charge state q and the extraction potential difference $\Delta\Phi$, and the magnetic flux density must be chosen according to the desired $m/q$ ratio (but, at the same time operating the ion source with an optimized **B**). In the part of Figure 9 showing the simulation region, the magnetic field is set for $Ar^{2+}$ for the given extraction voltage. The magnetic field is sufficiently strong that low energy electrons are not transported across the extraction; inside the dipole field, electrons can only gyrate along magnetic field lines.

Experimentally, ion currents up to 100 electrical mA (emA, e.g., Argon) at an extraction voltage of 15 kV could be extracted and transported. Such a beam would be rapidly lost (by SC blow-up) due to its own SC if there were no SCC. However, the electrons are of low energy and oscillate around field lines in the extracted ion beam.

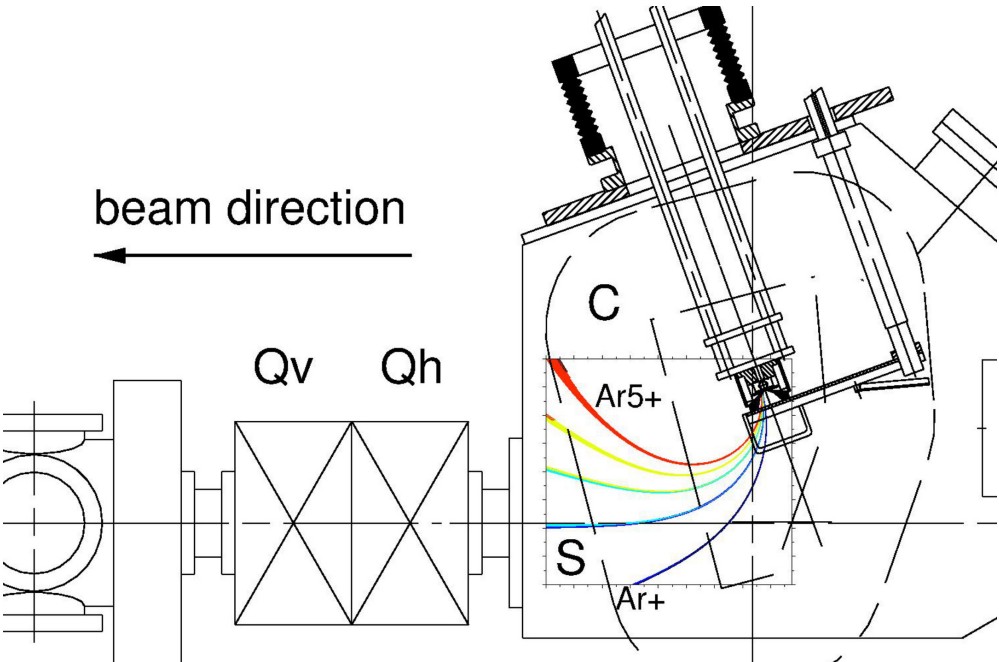

**Figure 9.** PIG ion source with radial ion beam extraction within the magnetic dipole field perpendicular to the drawing plane. Different charge states from Argon ($Ar^+$ to $Ar^{5+}$) will be extracted and deflected according to their $m/q$ ratio. The beam transport through that magnetic dipole field produced by the coil **C** would not be possible without a high degree of SCC. Two magnetic quadrupoles behind the dipole focus the ion beam in horizontal direction **Qh** and in vertical direction **Qv**, respectively. Simulation area is marked with S.

Now, returning to the extracted ion beam from CAPRICE ECRIS, and with the assumption of SCC even within a magnetic dipole, the profile on a VT of the $m/q$-analyzed beam behind the dipole looks even more complicated, but no publication seems to have addressed this to explain such experimental result, as shown in Figure 6. The guess seems plausible that the visible structure on the ion beam is already stamped by the plasma density distribution along the radial loss lines within the ion source. Ions having the most skewed trajectories do have the lowest longitudinal velocity component, resulting in different focusing properties by the magnetic solenoid. With that assumption the results in Figures 5 and 6 can be explained.

One might conclude that SCC has to build up locally in the different beam line sections and that the ion beam tends to compensate itself over time so long as no active removal of electrons occurs. The necessary time to establish this SCC can be measured best by a pulsed operation of the ion source or by creating a pulsed ion beam out of the dc ion beam using a chopper: two metallic plates are connected to a bipolar dc voltage power supply and then grounded by a fast switch. When the chopper plates are active, the resulting **E**-field displaces the beam off axis, and when they are grounded the beam returns to the axis and the trapping of electrons for SCC commences.

The result of such an experiment is shown in Figure 10. An ion beam extracted from an ECRIS is filtered by a magnetic dipole to select a specific $m/q$ (here $^4He^+$), followed by a chopper, a profile grid, and a Faraday cup (FC) to collect all ions with radii less than the FC radius. In front of the chopper the beam intensity is constant; behind the chopper the beam is pulsed. At the beginning of the ion beam pulse not enough electrons are available to achieve SCC of the positive ion beam. The beam is too divergent to totally fit into the

FC. With time, enough electrons are trapped in the ion beam, and more ions will fit into the FC as the beam divergence decreases. The necessary time for SCC can be decreased by creating more electrons. Increasing the pressure in the vacuum chamber is one way. Another possible electron source might be, e.g., a profile grid which is positioned in the beam path. Due to the limited transmission of such a grid, defined by the wire diameter and the number of wires, the intensity drops by this transmission factor, but the required time for SCC decreases from 3 ms to several 10 µs, indicating increased electron production rate. Beam line pressure is in the $10^{-8}$ mbar range.

This time dependency and its general SCC behavior is independent of the specific ion source; any beam to be transported will show similar behavior. No differences have been observed for noisy beams or more quiescent beams. In Figure 11, a Uranium ion beam from a MEVVA ion source is transported from the source toward an rf-accelerator over a 12 m long transport beam line consisting of drift spaces and magnetic lenses (dipole, quadrupoles). As a diagnostic, four beam transformers show the same ion pulse at different locations along the beam line. Beam line pressure is in the $10^{-7}$ mbar range.

The first BT shows all charge states simultaneously with electrical current 24 emA. The second BT indicates 10 emA, mainly $^{238}U^{4+}$. The next BT shows 8 emA, and behind a chopper [20], selecting a part of the beam, the fourth BT shows 4 emA. With increasing path length the time necessary to attain adequate SCC increases; this makes the chopper plausible, so as not to overload the rf-structure with unmatched beam. The ion beam, drifting from the chopper toward the 4th beam transformer, again requires time to restart the build up of SCC.

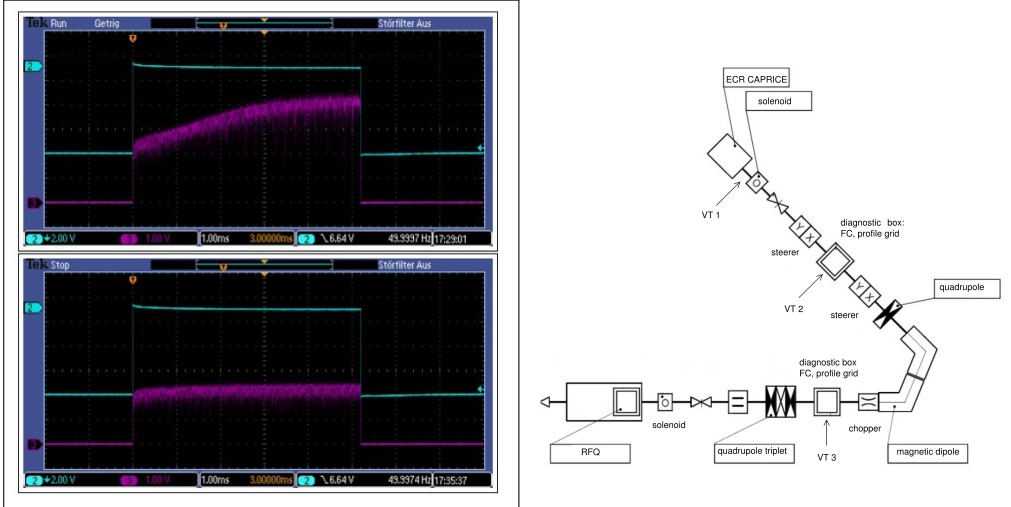

**Figure 10.** (**Left**): Pulse shape of a chopped ion beam, measured with a Faraday cup (FC) (violet) and timing signal for the chopper (blue). (**Top**): grid removed from the beam, (**Bottom**): profile grid inside the beam path. (**Right**): layout of beamline, showing the location of the chopper followed by the diagnostic box containing FC and profile grid. The location of each viewing target (VT 1–3) is shown also in experiments performed at GSI Darmstadt, Germany.

Optical elements designed for the primary ion beam will have a strong influence on the behavior of secondaries along the beam line:

- In field-free regions, electrons can move in any direction (longitudinal, transverse). An electric plasma and SCC will develop with time.
- Electric devices such as bends, Einzel lenses, quadrupoles, or similar will separate the ions and electrons in the beam, and space charge compensation will be lost, at least partially within the specific element.
- Magnetic devices such as dipoles, solenoids, quadrupoles, or similar will magnetize the compensating particles. SCC will take place along magnetic field lines.

- Along the beam line, the ion beam will experience different sections, separated from each other, where the actual SCC can behave differently. Plasma sheets might appear on the interfaces between different sections.

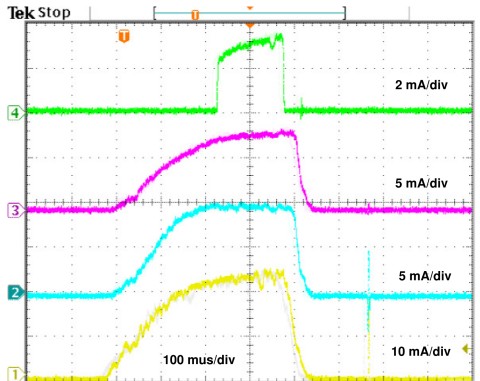 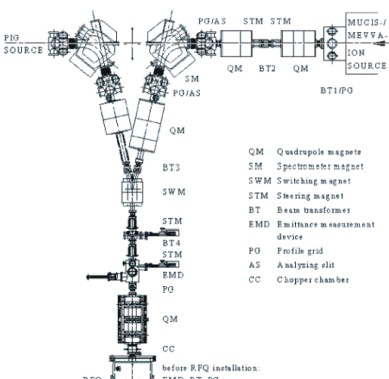

**Figure 11.** (**Left**): Signals from four beam transformers showing the same ion beam pulse along a 12 m long beam line. Behind a chopper SCC which needs to be rebuilt, requiring additional time. (**Right**): beam line from acceleration gap to the RFQ. QM magnetic quadrupole, SM spectrometer magnet, SWM swiching magnet (between beam lines), BT beam transformer.

## 5. Conclusions

Plasma produced within an ion source might have properties which could influence the extractable ion beam for the desired application. For ECRISs, the extracted ions seem to originate close to the location of the radial loss lines. The ions are then guided by the magnetic field lines crossing the extraction aperture, which explains the typical structure of any ECRIS ion beam. A simulation procedure is given which reproduces these typical experimental results. The specific ion beam profile behind any ECRIS can be explained with different longitudinal velocities for ions with different angles after extraction.

SCC is not a property of the ion source itself but a property of the beam line. With proper design of the beam line, and on time scales longer than the build-up times for SCC, beam transport can profit from SCC.

**Funding:** This research received no external funding.

**Institutional Review Board Statement:** Not applicable.

**Informed Consent Statement:** Not applicable.

**Data Availability Statement:** Not applicable.

**Acknowledgments:** The author would like to thank the responsible people at GSI, N. Angert, B.H. Wolf, and B. Franzke, who provided the opportunity to perform the experiments and computer simulations. Questions and discussions were always guided by the goal of improving the accelerator at GSI, which included the ion source and low energy beam transport.

**Conflicts of Interest:** The authors declare no conflict of interest.

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
