# Peer review of "An Ion Source’s View of Its Plasma"

_plasma, doi:10.3390/plasma4020023_

Round 1

Reviewer 1 Report

The paper deals with a deep insight of the extraction process in the ECR ion sources. A model of this process along with a code has been developed in the years to model this process which plays a key role in the quality of the beam produced.

Paper deals also with the effect of not-uniformity of the beam  extracted due to the variation of pumping frequency which is typical in this kind of sources affecting the starting conditions of the trajectories.

Very interesting the beam transport section with the consideration of SCC along a 12 m long line transporting high intensity of U4+.

A figure to understand position of the chopper and position of beam transformers will certainly help the readers. 

Author Response

Dear reviewer,

thank you very much for your review. I tried improve my English with a help of an American friend (US language, SF) and the Word spellchecker.

I also included the beam line figures to identify specific location of different beam line elements.

Reviewer 2 Report

The article provides an overview of the physics of ion beam extraction from Electron Cyclotron Resonance Ion Sources. The questions raised in the paper are scientifically sound and important for the relevant community.

My main objective to the paper in its current form is the general lack of clarity and structure. Experimental conditions and experimental results are not described in sufficient detail (e.g. spatial dimensions, plasma parameters, values of electrical and magnetic fields, definition of spatial axes). The theory is described in a "hand waving" and inaccurate manner. The simulation software used in this work is not described and the simulation framework is not presented. The comparison between the experiment and the respective theoretical predictions is not obvious for an average reader who is not familiar with the specific experimental set up.

Specific comments

  1. Terms "plasma temperature" and "plasma pressure" are used throughout the paper. In nonequilibrium plasmas, different species are characterized by different temperatures and pressures.
  2. Terms like "too high", "very thin", "more triangular", "fine enough", "some depth" should be avoided.
  3. 2nd paragraph on page 4 is unclear and should be improved
  4. Penultimate paragraph on page 5 is unclear
  5. Last paragpraph on page 5. What kind of collisions are effective in relevant plasma conditions? Collision frequency and mean free path should be estimated. 
  6. Last paragraph on page 6 is not clear. The photos in Figure 3 are intended to support the theoretical predictions, while 3 alternative interpretations of the erosion shape are proposed.
  7. Last 2 paragraphs on page 8 are not clear. What exactly are the 2 effects?
  8. Figure 6 caption is not clear. What is plotted in the figure?

Author Response

Dear reviewer,

thank you very much for your review.

I tried to improve the English with an American friend (US, SF) and using the Word spell checker.

The question which software has been used was specified (kobra3-inp, line 203, ref.10). Normally, my name is worldwide associated with kobra3-inp. I do present a recipe to handle the ecris extraction and its physical reasoning for it.

I am sorry that this sounds handwaving to you. But, as an ion source man I tried to describe the problem mainly from experimental facts, comparing them with simulation results based on theoretical assumptions. 

This might explain partially my way of writing in the paper. It also should encourage us all to a closer collaboration of experimentalists and theoreticans.

And, what is important to me as well: The fact, that I am able to reproduce experimental results with KOBRA3-INP this is NOT a proof, maybe an indication. This was also in my mind when listing the possible reasons - and not to exclude them from the beginning...

Let me try to respond to your individual points:

  1. I agree. The specific issue you mentioned would specify different temperatures for different isotopes and charge states. In the present paper I deal with the question of initial origin of the extracted ions. That is one step before looking for the different initial conditions of different elements/charge state.
  2. I agree again. Absolute dependencies should be preferred, but for the description of general behavior relative dependencies could be useful as well, e.g., at low pressure there is no discharge, when raising the pressure something will happen... This is already a physical effect without mention that the base pressure is x mBar. I tried to describe some effects (like the development of the profile with increasing focusing strength of a lense).
  3. Fig. 5 is a good example, the effect is understandable even without knowledge at how much [T] of the solenoid and [V] of the extraction system it happens.
  4. In this chapter I tried to summarize different parameters which influence the plasma. There are more parameters  than B and omega having influence. All dependencies need to be included in a simulation model. Again, I started with the location of an ion, starting velocity will be dependent on the temperature, of course, and the direction of this velocity is important. However, such dependencies are not yet considered in the paper.
  5. This is the same issue as in the previous point. For the simulation initial values are required to solve the equation of motion. Either a plasma simulator is available to determine these initial conditions - or other estimations are required.
  6. Collision frequency and mean free path length are well defined for electric plasma. It becomes more complicated in the presence of B.
  7. Figure 3 shows an experimental result. I think it is legitimate to put the question on the table what the possible reasons are for this experimental result. I tried to summarize all possible explanations. Of course I do have an opinion about the correct answer, but my aim was not to exclude any possible answer from a discussion. These plasma erosion traces are known for decades, but I could not find any explanation for them in the literature as well as at conferences or workshops. Only experimentalists do know these marks and on questions about there origin, a smiling face is normally the answer.
  8. I tried to list these two different experimental results: 1) the profile looks the same for each m/q. This sounds trivial, but at some conferences the effect of space charge were assumed to be the reason for the strange profile (lower m/q will be focussed stronger than larger m/q for the same extraction voltage). But if even m/q=1 does have this profile this CANNOT be the reason...             

The other effect is the behavior of focussing of each specific m/q for small steps of the focussing strebgth. From that effect it is reasonable that ions having the strongest angles do have the lowest longitudinal velocity (and this will effect the focussing). I tried to improve the figure caption.

Reviewer 3 Report

Review to the paper of Peter Spaedke,

The peer-reviewed article is written by a highly qualified researcher in the field of ion sources for heavy ion accelerators. For many years he headed the group of ion sources at the GSE accelerator center, Darmstadt, Germany.

The author of the article is well known in the world scientific and technical community, primarily as the developer of the Cobra software codes for modeling the formation of ion beam transport in the primary acceleration region of ion sources for various purposes, mainly ion sources for ion accelerator injectors. The experience of using these codes for solving specific problems in the development of ion accelerator technology is reflected in this article.

This article is of interest to developers of ion sources and may be published in the journal Plasma.

A small note: the name of the Russian scientist mentioned in link 20, Boris Makov, should be corrected.

Author Response

Dear reviewer,

Many thanks for your nice review! Of course, I did correct the spelling for Boris.

Many thanks again

Peter Spädtke

Round 2

Reviewer 2 Report

The Author has fully addressed all questions raised in the first review.